# Learning Effective Dynamics across Spatio-Temporal Scales of Complex Flows

Han Gao[1], Sebastian Kaltenbach [1], Petros Koumoutsakos [1*]
[1]Harvard SEAS
{hgao1,skaltenbach,petros}@seas.harvard.edu

Modeling and simulation of complex fluid flows with dynamics that span multiple spatio-temporal scales is a fundamental challenge in many scientific and engineering domains. Full-scale resolving simulations for systems such as highly turbulent flows are not feasible in the foreseeable future, and reduced-order models must capture dynamics that involve interactions across scales. In the present work, we propose a novel framework, Graph-based Learning of Effective Dynamics (Graph-LED), that leverages graph neural networks (GNNs), as well as an attention-based autoregressive model, to extract the effective dynamics from a small amount of simulation data. GNNs represent flow fields on unstructured meshes as graphs and effectively handle complex geometries and non-uniform grids. The proposed method combines a GNN based, dimensionality reduction for variable-size unstructured meshes with an autoregressive temporal attention model that can learn temporal dependencies automatically. We evaluated the proposed approach on a suite of fluid dynamics problems, including flow past a cylinder and flow over a backward-facing step over a range of Reynolds numbers. The results demonstrate robust and effective forecasting of spatio-temporal physics; in the case of the flow past a cylinder, both small-scale effects that occur close to the cylinder as well as its wake are accurately captured.

## 1. Introduction

Simulating complex systems that exhibit dynamics across multiple spatial and temporal scales remains a key challenge in a wide array of scientific and engineering disciplines. From turbulence [1] and climate modeling [2] to ocean dynamics [3] and biological systems [4], accurately capturing the intricate interplay between various scales is essential to understand, predict, and optimize system behavior. Traditional high-fidelity simulations, while accurate, often require substantial computational resources, rendering them impractical for real-time applications or extensive parametric studies. Consequently, reduced-order modeling (ROM) has emerged as a vital strategy for simplifying these complex systems by constructing lower-dimensional representations that preserve essential dynamics.

Despite its promise, ROM faces significant hurdles, particularly in scenarios where multiple regimes or scales interact within a single system [5]. Capturing nuanced interactions between different spatial and temporal scales in a unified reduced-order framework requires sophisticated modeling techniques that can adapt to varying geometries, resolutions, and dynamic behaviors. Traditional ROM approaches, such as Proper Orthogonal Decomposition (POD) [6] or Galerkin projections [7, 8], often struggle with scalability and flexibility when dealing with unstructured meshes or non-linear dynamics inherent in multi-scale systems.

Graph neural networks (GNNs) present substantial advantages for the modeling of discretized spatiotemporal systems governed by partial differential equations (PDEs) within unstructured meshes. These advantages encompass efficient spatial computation allocation, geometric adaptability, and end-to-end learning capabilities [9–15]. Nonetheless, the precision of GNN models often dimin-

---

*Corresponding author

Second Conference on Parsimony and Learning (CPAL 2025).

ishes when confronted with discontinuities and oscillations within flow fields, resulting in a decline in performance. These nonlinear instabilities become increasingly pronounced with extended rollout spans [16, 17]. Therefore, formulating robust and precise strategies to stabilize GNNs under such challenging flow scenarios is essential to unlocking their potential for real-world applications.

Several approaches have been proposed to stabilize rollouts, most of which are refinements of teacher-forcing methods [18], incorporating specialized training strategies to mitigate error accumulation in long-span rollouts. One common technique is noise injection, which augments the training dataset with perturbed samples to improve the model's robustness against noise. This approach is widely adopted for learning complex flows [9–11]. Another method involves feature conditioning, where additional neural networks with tailored input features enhance the expressive power of GNNs [12]. Although this method improves rollout accuracy, it increases computational cost and lacks clear guidelines for selecting features in complex problems. A variant of teacher-forcing uses multiple previous steps as input, enabling the model to use a longer history for more accurate predictions [9, 16]. Recent studies have demonstrated the benefits of optimizing the number of previous steps for improved accuracy [9]. However, this approach is computationally expensive, as it requires direct operations on a large number of nodes per step, leading to significant memory demands during gradient calculations and storage. A fundamental limitation of these teacher-forcing methods is the inconsistency between training and testing. Training relies on one-step predictions, whereas testing involves autoregressive multistep rollouts, exacerbating error accumulation and limiting long-term stability.

Sequence neural networks have recently emerged as powerful tools for simulating dynamical systems governed by PDEs, including long- and short-term memory (LSTM) networks [19, 20] and attention-based models [21]. Similar to teacher-forcing methods, these models use previous states to predict future ones. However, unlike teacher-forcing, sequence models do not rely on a small number of past steps for predictions. Instead, they leverage long-span dependencies by incorporating unique mechanisms, such as LSTM cells and hidden states [22] or attention mechanisms [23], enabling more general and robust forecasting. Although these innovations overcome the limitations of short-span dependencies, they often require significant memory to store long-term spatiotemporal sequences. Consequently, dimension reduction techniques are commonly employed to ensure efficient and effective training [24–33]. For example, sequential networks combined with dimension reduction methods, such as convolutional neural networks or proper orthogonal decomposition, have been successfully applied to simulate unstable, convection dominated and reacting flows of varying complexity [20, 21, 34–38]. Despite their promise, there remains a lack of robust frameworks that integrate GNNs with sequential learning architectures to handle unstructured flow data, particularly for moving or variable-size meshes. This work addresses this gap.

In this work, we propose a novel learning architecture that integrates a GNN-based autoencoder with a temporal attention model to efficiently predict spatio-temporal physical systems discretized on unstructured meshes. While GNNs have already been successfully employed to encode data or parameters in the context of simulations [39–42], using the encoded structure to forecast the evolution of high-dimensional, multi-scale systems of interest has not yet been explored. The Graph-LED framework comprises two main components: spatial dimension reduction and temporal forecasting. Spatial dimension reduction is achieved using a mesh-based GNN encoder-decoder, which aggregates local information of solution fields into node-level representations through multiple GNN layers. Our approach uses GNNs, enabling effective dimension reduction and recovery while preserving mesh information. Moreover, regions with different information and node density are readily incorporated into the framework. Temporal prediction is performed using an attention mechanism that facilitates dynamic learning and forecasting in a consistent autoregressive manner, eliminating the need for noise injection to stabilize rollouts. The proposed framework is validated on two incompressible flow scenarios with complex mesh configurations: flow past a cylinder at a Reynolds number of 696 and flow over a backward-facing step at $Re = 5000$. These cases feature multiscale flow characteristics, demonstrating the framework's ability to effectively handle intricate unstructured meshes and capture the complex dynamics of challenging physical systems.

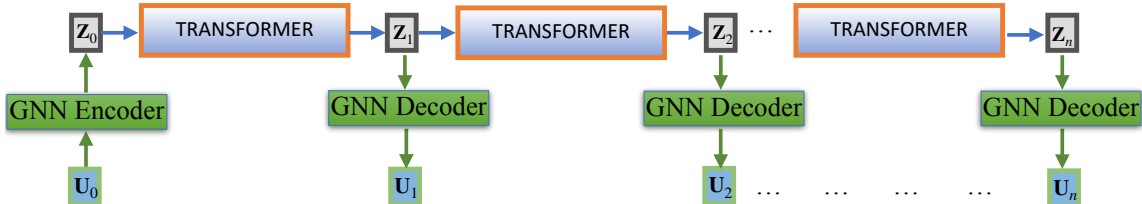

Figure 1: Overview of Graph-LED: A high-dimensional initial state $U_0$ is mapped to a low-dimensional latent representation $Z_0$ via a GNN Encoder. Subsequently this latent representation is propagated for $n$ steps using a Transformer. The GNN Decoder then maps the forecasted low-dimensional state $Z_n$ back to the high-dimensional state of interest $U_n$.

# 2. Methodology

In this section, we are presenting the Graph-LED framework. An overview of the framework can be found in Figure 1. Afterwards, we are introducing the high-dimensional nonlinear system of interest in Section 2.1. Section 2.2 shows how the state of such a system can be naturally represented by a graph. Section 2.3 explains the Graph-LED framework in detail, including the architecture choices (Section 2.3.1), the interpolation tool (Section 2.3.2) and the dimension reduction via GNN (Section 2.3.3). Section 2.3.4 details the temporal modeling in the low dimensional space.

## 2.1. Navier-Stokes Equations

In this section, we define the governing PDEs discussed throughout this paper. We note that while within this paper we target fluid dynamics as our application, the framework presented can also be applied to other PDEs of interest.

Let $\Omega \subset \mathbb{R}^2$ denote the spatial domain and $\mathcal{I} \equiv (0, \tau]$ the time domain, where $\tau \in \mathbb{R}_{>0}$ is the end time. For a two-dimensional Cartesian coordinate system, the spatial coordinate is $\mathbf{x} = [x, y]$, and the spatial gradient operator is $\nabla(*) := \left[ \frac{\partial(*)}{\partial x}, \frac{\partial(*)}{\partial y} \right]$. Here, $\mathbf{v} : \Omega \times \mathcal{I} \to \mathbb{R}^2$ denotes the velocity vector, typically expressed as $\mathbf{v} = [u, v]$. Subsequently, the time-dependent Navier-Stokes (NS) equations are given by:

$$\frac{\partial \rho}{\partial t} + \nabla \cdot (\rho \mathbf{v}) = 0, \quad \text{in } \Omega \times \mathcal{I}, \tag{1}$$

and

$$\frac{\partial}{\partial t} (\rho \mathbf{v}) + \nabla \cdot (\rho \mathbf{v} \mathbf{v}) = \nabla \cdot (\mu \nabla \mathbf{v}) - \nabla p + \nabla \cdot \left( \mu \nabla \mathbf{v}^{\mathrm{T}} \right) - \frac{2}{3} \nabla (\mu \nabla \cdot \mathbf{v}), \quad \text{in } \Omega \times \mathcal{I}, \tag{2}$$

where $\rho : \Omega \times \mathcal{I} \to 1$ is the constant normalized density, $p : \Omega \times \mathcal{I} \to \mathbb{R}$ is the pressure, and $\mu$ is the viscosity.

The vorticity $w : \Omega \times \mathcal{I} \to \mathbb{R}$ of the flow field can be computed as

$$w := \frac{\partial v}{\partial x} - \frac{\partial u}{\partial y} \tag{3}$$

The Reynolds number, which is often used to characterize a fluid flow, is defined as:

$$Re := \frac{\rho |U| l_{\text{ref}}}{\mu}, \tag{4}$$

where $l_{\text{ref}}$ is the reference length and $U$ the flow speed.

## 2.2. Discretization and graph representation

To compute approximate solutions of the NS equations presented above, we use the finite-volume (FV) method for spatial discretization [43]. Let $\mathcal{E}$ denote an unstructured (flexible grid with irregularly arranged elements) FV mesh, defined as a collection of non-overlapping cells that cover

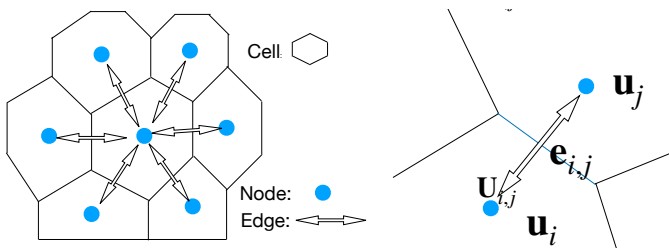

Figure 2: Left: Visualization of a FV mesh that can be naturally represented by a Graph. Right: Definition of graph nodes $\boldsymbol{u}_j$ as cells and graph edges $\boldsymbol{e}_{ij}$ as connection between two adjacent cells

the domain $\Omega$. These cells are ordered as $\mathcal{E} = \{\Omega_1, \ldots, \Omega_N\}$. For any two adjacent cells, $\Omega_i$ and $\Omega_j$, their shared metrics, such as relative displacement and the area of their shared boundary, are represented by a characteristic vector $\mathbf{e}_{i,j} \in \mathbb{R}^{N_\mathbf{e}}$. At any given time, each cell $\Omega_i$ is associated with a solution vector $\mathbf{u}_i \in \mathbb{R}^{N_\mathbf{u}}$, which approximates the continuous solution evaluated at the center of $\Omega_i$. As shown in Fig. 2, we can use an undirected graph to describe a snapshot of a mesh-based solution as

$$G = (\mathbf{U}, \mathbf{A}, E). \tag{5}$$

$\mathbf{U} = [\mathbf{u}_1, \ldots, \mathbf{u}_N] \in \mathbb{R}^{N_u \times N}$ is the vector of features of the node of the graph where subscripts represent nodes. $\mathbf{A} \in \mathbb{R}^{N \times N}$ is the graph adjacency matrix with binary entries, if $(\mathbf{A})_{i,j} = 1$, there is an edge connection between node $i$ and node $j$, which indicates that cell $\Omega_i$, $\Omega_j$ are adjacent to each other. We can obtain the total number of edges via the adjacency matrix as $N_E = \sum_i^N \sum_j^i (\mathbf{A})_{i,j}$. $E = \{\mathbf{e}_{i,j} | (\mathbf{A})_{i,j} = 1\}$ stores all the edge features of $N_E$.

**Remark 1** *In the context of graph representation, the edge is an abstract concept, not a geometric object. We call two nodes in a graph connected via an edge. However, in a mesh, an edge is often referred to as the line connecting two vertices. Furthermore, graph representation is not only applicable to FV meshes, it can also represent other mesh formats used for finite element or finite difference.*

We apply a numerical integral method (e.g., Euler or Runge-Kutta) together with the spatial discretization of FV [44]. Given time points of interest $\mathcal{T} = \{t_0, t_1, \ldots, t_{N_t} \mid t_{j+1} - t_j = \Delta t \quad j = 0, \ldots, N_t - 1\} \subset \mathcal{I}$ and initial condition $\mathbf{U}_0^*$, the sequence of the solution is

$$\mathcal{U}^F = \{\mathbf{U}_1^*, \ldots, \mathbf{U}_{N_t}^* \mid \mathbf{U}_{j+1}^* = \mathcal{F}(\mathbf{U}_j^*; \Delta t, \delta t) \quad j = 0, \ldots, N_t - 1\}, \tag{6}$$

where $j$ is for $t_j$. And $\mathcal{F} : \mathbb{R}^{N_u \times N} \times \mathbb{R} \times \mathbb{R} \to \mathbb{R}^{N_u \times N}$ is the forward solver to advance the solution to the next time step of interest, where $\delta t$ is the actual numerical integral step.

To resolve all scales of interest, $N$ can be very large [43]. To save computational costs and facilitate many-query tasks, we seek to approximate $\mathcal{U}_i^F \approx \mathcal{U}_i^{F_r}$, which will be detailed in the following sections.

## 2.3. Learning effective dynamics across spatiotemporal scales

Our framework consists of two main parts: We initially employ a GNN as an encoder-decoder framework, which transforms high-dimensional states into a low-dimensional latent space, as elaborated in Section 2.3.1. Subsequently, the encoder and decoder derived are used to train a transformer model to predict dynamics within low-dimensional space, as discussed in Section 2.3.4.

### 2.3.1. GNN architecture

We utilize multilayer perceptrons (MLP), which are fully connected feedforward neural networks. These networks facilitate the mapping of vectors from dimension $N_{\text{in}}$ to $N_{\text{out}}$, denoted as $\text{mlp} : \mathbb{R}^{N_{\text{in}}} \to \mathbb{R}^{N_{\text{out}}}$ and $\mathbf{u} \mapsto \text{mlp}(\mathbf{u})$. Layer normalization (LNM) is employed to perform an affine

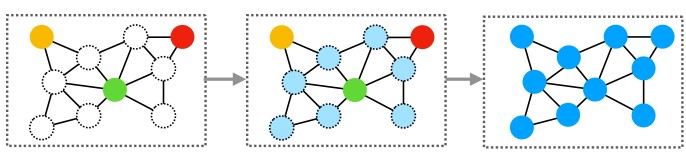

Figure 3: Interpolation to new coordinates: Based on the three colored nodes, we reconstruct the node values at new nodes via nearest neighbor interpolation, i.e. weighting the contribution of each colored node according to their distance from the new node. Finally we are able to construct a new larger or smaller graph.

transformation on a vector $\mathbf{u}$ using learnable parameters [45], indicated as $\mathrm{lnm} : \mathbb{R}^{N_{\mathrm{in}}} \to \mathbb{R}^{N_{\mathrm{in}}}$ and $\mathbf{u} \mapsto \mathrm{lnm}(\mathbf{u})$. Subsequently, we present the GNN layer as described in (7), which is adapted from [9]. This layer processes an input graph ($G_1$) and produces an output graph ($G_2$): $G_1 \mapsto G_2 = \mathrm{gnn}(G_1)$. Within a graph, any given node consolidates the features of its neighboring nodes in a two-step process. The first step is to update the edge feature vector: an MLP ($\mathrm{mlp_e}$) takes the input that concatenates the center node feature ($\mathbf{u}_i^1$, where the superscript shows to which graph it belongs to), the neighbor nodes' ($j \in \mathcal{N}(i)$) and edge features ($\mathbf{u}_j^1, \mathbf{e}_{i,j}^1$), and outputs the new edge feature ($\mathbf{e}_{i,j}^2$). The second step is to update the node feature vector. Another MLP ($\mathrm{mlp_n}$) takes the input, which concatenates the central node feature ($\mathbf{u}_i^1$), and the mean updated edge features of all neighbors ($\frac{1}{|\mathcal{N}(i)|} \sum_{j \in \mathcal{N}(i)} \mathbf{e}_{i,j}^2$) to update the center node feature $\mathbf{u}_i^2$. In addition, the residual connection and layer normalization ($\mathrm{lnm_e}, \mathrm{lnm_n}$) are applied for both steps. These two steps can be expressed by the function of a GNN layer: $G_1 = (\mathbf{U}_1, \mathbf{A}, E_1), G_2 = (\mathbf{U}_2, \mathbf{A}, E_2), \mathrm{gnn} : G_1 \mapsto G_2$,

$$
\begin{aligned}
E_1 &= \{\mathbf{e}_{i,j}^1\}, \quad \mathbf{U}_1 = [\mathbf{u}_1^1, \dots, \mathbf{u}_N^1], \\
E_2 &= \{\mathbf{e}_{i,j}^2 \mid \mathbf{e}_{i,j}^2 = \mathbf{e}_{i,j}^1 + \mathrm{lnm_e} \circ \mathrm{mlp_e}([\mathbf{u}_i^1, \mathbf{u}_j^1, \mathbf{e}_{i,j}^1])\}, \\
\mathbf{U}_2 &= \left[\mathbf{u}_1^2, \dots, \mathbf{u}_N^2 \middle| \mathbf{u}_i^2 = \mathbf{u}_i^1 + \mathrm{lnm_n} \circ \mathrm{mlp_n}([\mathbf{u}_i^1, \frac{1}{|\mathcal{N}(i)|} \sum_{j \in \mathcal{N}(i)} \mathbf{e}_{i,j}^2])\right].
\end{aligned}
\tag{7}
$$

$N_g$ GNN layers are stacked as a compound function $\mathrm{gnns}_{N_g} := \mathrm{gnn}_1 \circ \dots \circ \mathrm{gnn}_{N_g}$.

### 2.3.2. Coordinates assignment and interpolation

We associate every node in a graph $G$ (see Equation 5) with a coordinate, and $\mathcal{X}_1 = \{\mathbf{x}_1, ..., \mathbf{x}_{N_1}\}$ denotes the set of all node coordinates. For the graph feature vector $\mathbf{U} = [\mathbf{u}_1, \dots, \mathbf{u}_{|\mathcal{X}_1|}]$, we can perform a regular interpolation operation given a new set of coordinates $\mathcal{X}_2 = \{\mathbf{x}_1, ..., \mathbf{x}_{|\mathcal{X}_2|}\}$,

$$
\mathbf{Z} = [\mathbf{z}_1, \dots, \mathbf{z}_{|\mathcal{X}_2|}] = I_{\mathcal{X}_2}^{\mathcal{X}_1}(\mathbf{U}), \quad \mathbf{U}' = [\mathbf{u}_1', \dots, \mathbf{u}_{|\mathcal{X}_1|}'] = I_{\mathcal{X}_1}^{\mathcal{X}_2}(\mathbf{Z}),
\tag{8}
$$

where $I$ is a general interpolation method, $I_{\mathcal{X}_2}^{\mathcal{X}_1} : \mathbb{R}^{N_{\mathbf{u}} \times |\mathcal{X}_1|} \to \mathbb{R}^{N_{\mathbf{u}} \times |\mathcal{X}_2|}$ and $I_{\mathcal{X}_1}^{\mathcal{X}_2} : \mathbb{R}^{N_{\mathbf{u}} \times |\mathcal{X}_2|} \to \mathbb{R}^{N_{\mathbf{u}} \times |\mathcal{X}_1|}$. In this paper, we apply the nearest-neighbor interpolation method [46]. A visualization of this process during the up-sampling in the decoding can be found in Figure 3 and further explanation in Appendix B.

**Remark 2** *Graphs generally stay in the non-Euclidean space, and the node's position definition varies from problem to problem. The graph representation of meshes naturally provides the Euclidean coordinates of cells that can be used here to make GNNs mimic the encoder-decoder dimension reduction of CNNs.*

### 2.3.3. Dimension reduction and up-sampling via GNN

As all the components of the model have been introduced, we proceed to present the graph-mesh reducer designed for dimension reduction. We construct this encoder, as delineated in $\mathrm{Encoder} : (G_0, \mathcal{X}_1, \mathcal{X}_2, \mathbf{w}_{\mathrm{Encoder}}) \mapsto \mathbf{Z}$ within (9). The process begins with the application of an MLP ($\mathrm{mlp_0}, \mathrm{mlp_1}$) to transform node ($\mathbf{u}_i^0$) and edge features ($\mathbf{e}_{i,j}^0$) into their respective hidden features

$(\mathbf{U}_1, E_1)$, which are subsequently normalized using LNM $(\text{lnm}_0, \text{lnm}_1)$. These hidden node and edge features $(\mathbf{U}_1, E_1)$ constitute the hidden graph $(G_1)$. Thereafter, several GNN layers $(\text{gnn}_{N_g})$ are employed to derive an alternative hidden graph $(G_2)$. Once again, the MLP $(\text{mlp}_2)$ and LNM $(\text{lnm}_2)$ are utilized to manipulate the node feature $(\mathbf{u}_i^2)$ of $G_2$, resulting in the generation of a new feature vector $(\mathbf{U}_3)$. Given the two sets of coordinates, one $(\mathcal{X}_1)$ is associated with the input $\mathbf{U}_0$, and another $(\mathcal{X}_2)$ is associated with the output vector $\mathbf{Z}$. We perform interpolation to obtain the final reduced vector $\mathbf{Z}$.

$$
\begin{aligned}
G_0 &= \left(\mathbf{U}_0 = [\mathbf{u}_1^0, \ldots, \mathbf{u}_{|\mathcal{X}_1|}^0], \mathbf{A}, E_0 = \{\mathbf{e}_{i,j}^0\}\right), \\
\mathbf{U}_1 &= [\text{lnm}_0 \circ \text{mlp}_0(\mathbf{u}_1^0), \ldots, \text{lnm}_0 \circ \text{mlp}_0(\mathbf{u}_{|\mathcal{X}_1|}^0)], \\
E_1 &= \{\mathbf{e}_{i,j}^1 \mid \mathbf{e}_{i,j}^1 = \text{lnm}_1 \circ \text{mlp}_1(\mathbf{e}_{i,j}^0)\}, \\
G_1 &= (\mathbf{U}_1, \mathbf{A}, E_1), \\
G_2 &= \left(\mathbf{U}_2 = [\mathbf{u}_1^2, \ldots, \mathbf{u}_{|\mathcal{X}_1|}^2], \mathbf{A}, E_2 = \{\mathbf{e}_{i,j}^2\}\right) = \text{gnns}_{N_g}(G_1), \\
\mathbf{U}_3 &= [\text{lnm}_2 \circ \text{mlp}_2(\mathbf{u}_1^2), \ldots, \text{lnm}_2 \circ \text{mlp}_2(\mathbf{u}_{|\mathcal{X}_1|}^2)], \\
\mathbf{Z} &= [\mathbf{z}_1, ..., \mathbf{z}_{|\mathcal{X}_2|}] = \mathcal{I}_{\mathcal{X}_2}^{\mathcal{X}_1}(\mathbf{U}_3),
\end{aligned}
\tag{9}
$$

where $\mathbf{w}_{\text{Encoder}}$ are all trainable in the Encoder.

**Remark 3** *Using GNN layers, each node contains not only its information, but also its neighbors. Moreover, with a proper number of GNN layers, we could leave out a portion of the original nodes and only keep a small number of nodes $(|\mathcal{X}_2| \ll |\mathcal{X}_1|)$ which still contain the original graph information. The MLPs are flexible to control the dimension of the feature vector layer-wise. These are analogous to the concept of hidden channel number associated with height/width/length in CNNs.*

The graph-mesh up-sampler (Decoder) performs the reverse task to map a reduced vector $(\mathbf{Z})$ to a high-dimension vector $(\mathbf{U})$ ( see (10)). Given the coordinates $(\mathcal{X}_1, \mathcal{X}_2)$ associated with the output $(\mathbf{U})$ and the input $(\mathbf{Z})$. The hidden vector $(\mathbf{U}_0)$ is obtained by up-sampling $(\mathcal{I}_{\mathcal{X}_1}^{\mathcal{X}_2})$. Then another hidden vector $(\mathbf{U}_1)$ is processed by $\mathbf{U}_0$ through MLP $(\text{mlp}_0)$ and LNM $(\text{lnm}_0)$. Given the edge feature vectors $(E_0)$, we use MLP $(\text{mlp}_1)$ and LNM $(\text{lnm}_1)$ to obtain the hidden edge feature vectors $(E_1)$. Then $\mathbf{U}_1$ and $E_1$ compose the hidden graph $G_1$. After several GNN layers are obtained, $G_2$ is obtained, we use MLP $(\text{mlp}_2)$ to map the features of the hidden nodes $\mathbf{U}_2$ to the final output $\mathbf{U}$. These sequential steps can be summarized as Decoder : $(\mathbf{Z}, E_0 = \{\mathbf{e}_{i,j}^0\}, \mathcal{X}_1, \mathcal{X}_2, \mathbf{w}_{\text{Decoder}}) \mapsto \mathbf{U}$,

$$
\begin{aligned}
\mathbf{U}_0 &= [\mathbf{u}_1^0, \ldots, \mathbf{u}_{|\mathcal{X}_1|}^0] = \mathcal{I}_{\mathcal{X}_1}^{\mathcal{X}_2}(\mathbf{Z}), \\
\mathbf{U}_1 &= [\text{lnm}_0 \circ \text{mlp}_0(\mathbf{u}_1^0), \ldots, \text{lnm}_0 \circ \text{mlp}_0(\mathbf{u}_{|\mathcal{X}_1|}^0)], \\
E_1 &= \{\mathbf{e}_{i,j}^1 \mid \mathbf{e}_{i,j}^1 = \text{lnm}_1 \circ \text{mlp}_1(\mathbf{e}_{i,j}^0)\}, \\
G_1 &= (\mathbf{U}_1, \mathbf{A}, E_1), \\
G_2 &= \left(\mathbf{U}_2 = [\mathbf{u}_1^2, \ldots, \mathbf{u}_{|\mathcal{X}_1|}^2], \mathbf{A}, E_2 = \{\mathbf{e}_{i,j}^2\}\right) = \text{gnns}_{N_g}(G_1), \\
\mathbf{U} &= [\text{mlp}_2(\mathbf{u}_1^2), \ldots, \text{mlp}_2(\mathbf{u}_{|\mathcal{X}_1|}^2)],
\end{aligned}
\tag{10}
$$

where $\mathbf{w}_{\text{Decoder}}$ denotes all trainable parameters in the decoder.

**Remark 4** *In this study, a solution snapshot at time t constitutes a solution vector associated with a computational mesh. The mesh information can be used to assemble $E_0$ to facilitate graph neural networks (GNNs) in reconstructing this solution vector. Additionally, it is important to note that the coordinates are employed solely for interpolation purposes, thereby preserving the method's shift invariance. Both aspects resemble similarity to a convolutional neural network (CNN) decoder, which inherently exploits the structured mesh information provided.*

More details on training for the GNN encoder and decoder can be found in Appendix A.

### 2.3.4. Temporal model

The value, key, and query functions represent core components within an attention mechanism [23]. The value function facilitates the mapping of an initial vector to a resultant vector $\mathcal{V}: \mathbb{R}^{N_{\mathbf{z}}^1} \rightarrow$

$\mathbb{R}^{N_{\mathbf{Z}}^2}, \mathbf{Z} \mapsto \mathcal{V}(\mathbf{Z})$, where $N_{\mathbf{Z}}^1$ typically constitutes a $N_{\mathrm{h}}$-fold multiple of $N_{\mathbf{Z}}^2$. Meanwhile, the query and key functions are characterized by uniform input and output dimensionalities.

$$\begin{aligned}
\mathcal{Q} : \mathbb{R}^{N_{\mathbf{Z}}} \to \mathbb{R}^{N_{\mathbf{Z}}'}, \quad \mathbf{Z} \mapsto \mathcal{Q}(\mathbf{Z}), \\
\mathcal{K} : \mathbb{R}^{N_{\mathbf{Z}}} \to \mathbb{R}^{N_{\mathbf{Z}}'}, \quad \mathbf{Z} \mapsto \mathcal{K}(\mathbf{Z}).
\end{aligned} \tag{11}$$

Given a pair of query and key functions, the unnormalized attention function maps two vectors of the same dimension to a positive scalar $\mathcal{A} : \mathbb{R}^{N_{\mathbf{Z}}} \times \mathbb{R}^{N_{\mathbf{Z}}} \to \mathbb{R}_{>0}$,

$$\mathcal{A}(\mathbf{Z}_1, \mathbf{Z}_2) = \exp\left(\mathcal{Q}(\mathbf{Z}_1) \cdot \mathcal{K}(\mathbf{Z}_2)\right) = \exp\left(\sum_i \mathcal{Q}(\mathbf{Z}_1)_i \, \mathcal{K}(\mathbf{Z}_2)_i\right), \tag{12}$$

where $\mathbf{Z}_1$ is usually called the query vector and $\mathbf{Z}_2$ is the key vector.

For a sequence of vectors $\mathcal{S} = \{\mathbf{Z}_k\}_{k=1}^{N_{\mathrm{sl}}} \subset \mathbb{R}^{N_{\mathbf{Z}}}$, the $N_h$-head attention model maps any element vector of the sequence to a new vector with the same dimension denoted by $\mathrm{mhat}_{N_{\mathrm{h}}}^{\mathcal{S}} : \mathcal{S} \to \mathbb{R}^{N_{\mathbf{Z}}}$, $\forall \mathbf{Z}_i \in \mathcal{S}, \mathbf{Z}_i \mapsto \mathbf{V}_i$,

$$\begin{aligned}
a_{i,j}^{\mathrm{h}} &= \frac{\mathcal{A}^{\mathrm{h}}(\mathbf{Z}_i, \mathbf{Z}_j)}{\sum_{k=1}^{N_{\mathrm{sl}}} \mathcal{A}^{\mathrm{h}}(\mathbf{Z}_i, \mathbf{Z}_k)}, \quad \mathcal{A}^{\mathrm{h}} \in \{\mathcal{A}^1, \ldots, \mathcal{A}^{N_{\mathrm{h}}}\}, \quad \mathrm{h} = 1, \ldots, N_{\mathrm{h}}, \\
\mathbf{V}_i^{\mathrm{h}} &= \sum_{j=1}^{N_{\mathrm{sl}}} a_{i,j}^{\mathrm{h}} \mathcal{V}^{\mathrm{h}}(\mathbf{Z}_j), \quad \mathcal{V}^{\mathrm{h}} \in \{\mathcal{V}^1, \ldots, \mathcal{V}^{N_{\mathrm{h}}}\}, \quad \mathrm{h} = 1, \ldots, N_{\mathrm{h}}, \\
\mathbf{V}_i &= [\mathbf{V}_i^1, \ldots, \mathbf{V}_i^{N_{\mathrm{h}}}],
\end{aligned} \tag{13}$$

where $a_{i,j}^{\mathrm{h}}$ denotes the h-th head attention of vectors $\mathbf{Z}_i, \mathbf{Z}_j$, and $\mathbf{V}_i \in \mathbb{R}^{N_{\mathbf{Z}}}$ is output vector from the input vector $\mathbf{Z}_i$.

**Remark 5** *The formulation of query, key, and value functions is conceptually straightforward. These functions can derive from multi-layer perceptrons (MLPs) or alternative neural network constructs, provided that they maintain the vector dot-product operation [23]. Our multi-head attention model architecture adheres rigorously to the GPT-2 framework [47], which is appropriate for autoregressive predictions in dynamic systems.*

**Autoregressive, parametric sequence modeling**: The initial vector $(\mathbf{Z}_0)$ forms the starting set $(\mathcal{S}_0)$. The new reduced vector $(\mathbf{Z}_{j+1})$ is predicted by choosing the last current vector $(\mathbf{Z}_j)$ in the sequence $(\mathcal{S}_j)$ as input of the head attention model $N_{\mathrm{h}}$ $(\mathrm{mhat}_{N_{\mathrm{h}}}^{\mathcal{S}_j})$. We use a sliding window of length $N_{\mathrm{sw}}$ to keep the model only generating a new vector based on a constant number of previous vectors. Specifically, we drop out the older vectors outside the window, but always keep the parameter-dependent vector. This auto-regression can be mathematically expressed as $F_r : (\mathbf{Z}_0, \mathbf{w}_{F_r}) \to \{\mathbf{Z}_1, \ldots, \mathbf{Z}_{N_t}\}$,

$$\mathcal{S}_0 = \{\mathbf{Z}_0\},$$

$$\text{repeat} : \begin{cases}
\mathcal{S}_j = \begin{cases} \{\mathbf{Z}_0, \ldots, \mathbf{Z}_j\} & \text{if } j < N_{\mathrm{sw}} - 1, \\ \{\mathbf{Z}_{j+2-N_{\mathrm{sw}}}, \ldots, \mathbf{Z}_j\} & \text{if } j \geq N_{\mathrm{sw}} - 1, \end{cases} \\
\mathbf{Z}_{j+1} = \mathrm{mhat}_{N_{\mathrm{h}}}^{\mathcal{S}_j}(\mathbf{Z}_j), \\
\mathcal{S}_{j+1} = \begin{cases} \{\mathbf{Z}_0, \ldots, \mathbf{Z}_{j+1}\} & \text{if } j + 1 < N_{\mathrm{sw}} - 1, \\ \{\mathbf{Z}_{j+3-N_{\mathrm{sw}}}, \ldots, \mathbf{Z}_{j+1}\} & \text{if } j + 1 \geq N_{\mathrm{sw}} - 1, \end{cases}
\end{cases} \tag{14}$$

where $\mathbf{w}_{F_r}$ is the involved trainable weights of all the neural networks.

**Remark 6** *If the total length of interest for the prediction is short, the model can look back at all the previous vectors, leading the vector-vector attention matrix to be a lower triangular matrix; otherwise, the vector-vector attention matrix is a block diagonal matrix. The goal of setting the sliding window is to reduce computational costs due to the quadratic complexity of attention [23].*

Further details regarding the training for the introduced GNN encoder and decoder can be found in Section C.

# 3. Results

## 3.1. Cylinder flow

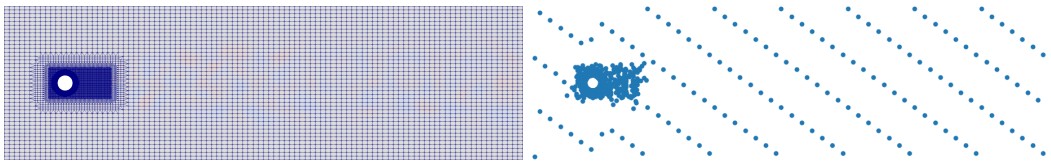

Figure 4: The system is reduced from 27,127 points to 1,024 points.

Our first test case is the 2-D flow past a cylinder at $Re = 696$ where the reference length is $l_{\mathrm{ref}} = 1$. In this work, our aim is to develop a surrogate model to predict the solution given any initial condition that lies in the solution manifold. Training (4500 snapshots) and testing (500 snapshots) data are generated using the OpenFOAM simulator [44], from which an unstructured mesh with different resolutions is used to discretize the domain (Fig. 4). The actual numerical integral step ($\delta t$) is 0.0025, and the Graph-LED is trained to directly evolve the system in a larger step of $\Delta t = 1$.

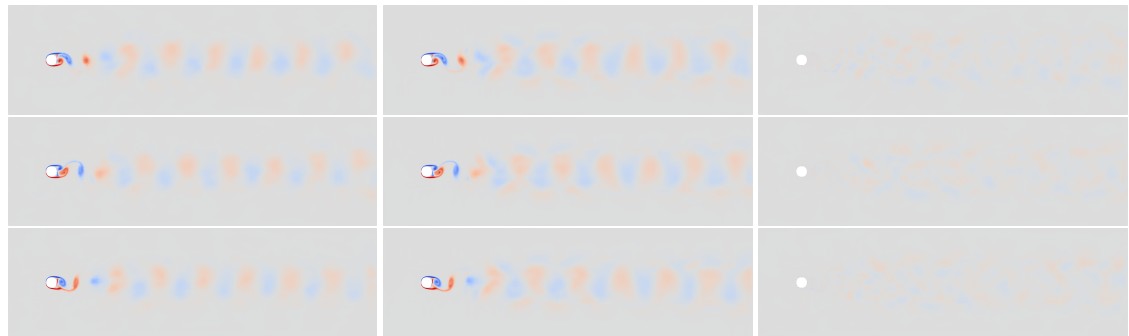

Figure 5: Vorticity ($\frac{\partial v}{\partial x} - \frac{\partial u}{\partial y}$) forecasted by Graph-LED (*left*), OpenFOAM (*middle*) and the error (*right*) from $t = 0, 50, 100$ (*from top row to bottom row*).

Given a testing initial condition, the predictions using the proposed Graph-LED and OpenFOAM are plotted in Fig. 5. The predicted vorticity fields are in good agreement with the OpenFOAM predictions but can be generated with a large speedup (900X). We note that the flow field in the wake is accuractly captured. When we examine the flow field near the cylinder (Fig. 8 in Appendix G), the pressure gradient over the cylinder and the wall shear stress (WSS) are well predicted by Graph-LED compared to OpenFOAM. Lastly, Graph-LED shows remarkable accuracy in capturing lift and drag coefficients (Fig. 9 in Appendix G), closely matching the results obtained from Open-FOAM, indicating that fine-scale high-frequency signals are effectively resolved. We compared the performance of Graph-LED with two other deep-learning baselines in Section F.

## 3.2. Flow over backward-facing step

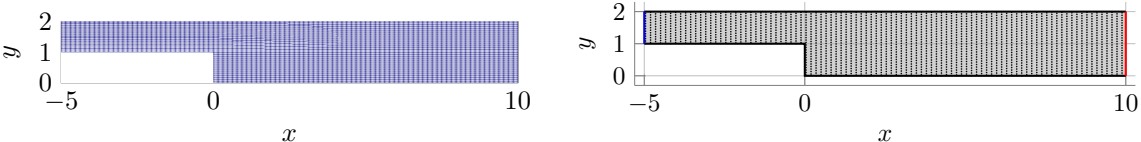

Figure 6: The system is reduced from 20,480 points to 2,048 points.

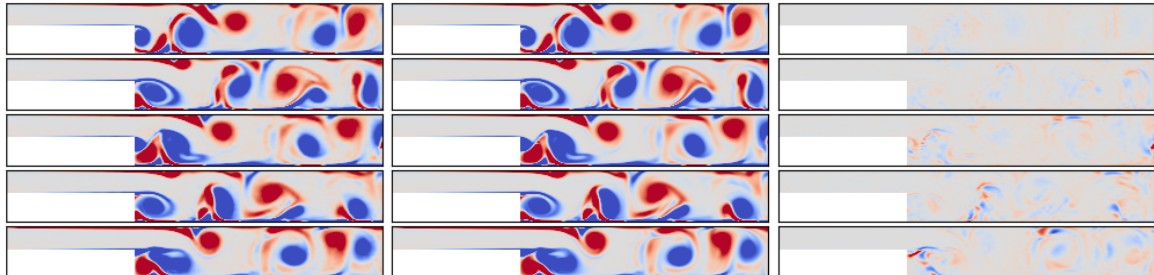

Figure 7: Vorticity ($\frac{\partial v}{\partial x} - \frac{\partial u}{\partial y}$) forecasted by Graph-LED (*left*), OpenFOAM (*middle*) and the error (*right*) from $t = 0, 0.5, 1, 1.5, 2$ (*from top row to bottom row*).

As a second case we consider the turbulent flow over a backward-facing step at $Re = 5000$, a classical problem in which previous work [48, 49] can only model the rectangular domain after the backward-facing step. However, we are interested in predicting not only the recirculation region after the step, but also the region in the pipe, that is, the entire domain (Fig. 6). The complete data set contains 10,000 snapshots with a time-step size of $\Delta t = 0.05$, where 5000 snapshots are used for training and the rest are used for testing. The integral time step for the numerical simulation with OpenFOAM is $\delta t = 2 \times 10^{-4}$.

Given an instant snapshot as an initial condition, the prediction of vorticity is plotted in Fig. 7. Graph-LED demonstrated reasonably good performance in forecasting the sequence over one flow-through time, achieving a significant speedup of around 100X.

Graph-LED displays an outstanding capability to predict the mean and variance of vorticity over time (Fig. 10 and Fig. 11 in the Appendix G), effectively capturing the intricate dynamics of the system with high precision. One of the most striking aspects of its predictive power lies in its ability to accurately represent the spatial evolution of the vorticity profile. Initially, at the leftmost part of the domain, the profile begins in a symmetric state. As the analysis extends across the spatial domain, the vorticity profile undergoes a gradual transformation, transitioning into an asymmetric structure in the middle regions. Subsequently, as the spatial location approaches the rightmost end of the domain, the profile regains its symmetry. This ability to trace and replicate the nuanced transitions between symmetry and asymmetry highlights Graph-LED's strength in modeling not only temporal changes but also spatial variations, offering a profound insight into the complex physical phenomena that govern the dynamics of the flow field.

## 4. Conclusions

We introduced an innovative framework that integrates GNNs with attention-based autoregressive models to address the complexities of modeling multiscale spatiotemporal dynamics in unstructured mesh data. By leveraging GNNs for spatial dimension reduction and temporal attention mechanisms for dynamic forecasting, the proposed method achieves significant improvements in accuracy, efficiency, and scalability. The framework's robustness was demonstrated through fluid dynamics scenarios such as flow past a cylinder and flow over a backward-facing step, where it accurately captured intricate details like small-scale effects and high-frequency signals while significantly reducing computational costs.

The findings highlight the potential of the Graph-LED approach to model and predict physical systems with multiple spatio-temporal scales of interest, bridging a critical gap in the simulation of multiscale systems. Future work could explore integrating adaptive refinement strategies and addressing limitations such as error propagation in longer-term predictions.

# Acknowledgements

S.K. and P.K. acknowledge support by the Defense Advanced Research Projects Agency (DARPA) through Award HR00112490489. H.G. and P.K. acknowledge support by the National Science Foundation (NSF) through Award CBET-2347423.

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

## A. GNN Architecture - Training Details

The success of Encoder and Decoder is inherently linked to the proper network weight vectors that ensure accurate recovery of the high-dimension vector $\mathbf{U}$. To this end, we let $\Xi = \{\boldsymbol{\mu}_1, \ldots, \boldsymbol{\mu}_{N_s}\} \subset \mathcal{D}$ be the collection of PDE encoder / encoder training parameters, and $G_{i,j}^* = (\mathbf{U}_{i,j}^*, \mathbf{A}_i, E_i)$ denotes the graph for $\boldsymbol{\mu}_i$ at time $t_j$ where $\mathbf{U}_{i,j}^*$ is from (6), $E_i, \mathbf{A}_i, \mathcal{X}_1^i, \mathcal{X}_2^i$ are from the mesh $\mathcal{E}_{\boldsymbol{\mu}_i}$ of $\boldsymbol{\mu}_i$. The optimal weights $\mathbf{w}_{\text{Encoder}}^*, \mathbf{w}_{\text{Decoder}}^*$ can be obtained via solving stochastic optimization with Adam method [50],

$$\underset{\mathbf{w}_{\text{Encoder}}, \mathbf{w}_{\text{Decoder}}}{\arg\min} \sum_{i=1}^{N_s} \sum_{j=1}^{N_t} \left\| \mathbf{U}_{i,j}^* - \text{Decoder}\Big( \big(\text{Encoder}(G_{i,j}^*, \mathcal{X}_1^i, \mathcal{X}_1^i, \mathbf{w}_{\text{Encoder}})\big), E_i, \mathcal{X}_i^1, \mathcal{X}_i^2, \mathbf{w}_{\text{Decoder}} \Big) \right\|_2. \tag{15}$$

**Remark 7** *For each scalar field in the training data, we use a normalization, where mean and variance are calculated from all of the training datasets.*

## B. Nearest-neighbor interpolation

We perform feature interpolation between the source and target point clouds in our framework [41]. This method leverages the search for the nearest neighbor k to propagate features $\mathbf{U}$, defined at the source points $\mathcal{X}_1$, to the target points $\mathcal{X}_2$, to obtain $\mathbf{Z}$. For each target point, feature values were calculated as the weighted average of features from its k-nearest neighbors among the source points, with weights inversely proportional to the Euclidean distances in the spatial domain. This interpolation method ensures smooth and spatially coherent feature transfer, enabling effective integration of geometric and feature information across scales in our model.

## C. Temporal model - Training Details

To mitigate the additional cost of training the dynamic model $(F_r)$, the training data are exactly the same as the data used to train the spatial models $(\text{Encoder}, \text{Decoder})$. To this end, we used the trained encoder $(\mathbf{w}_{\text{Encoder}}^*)$ to generate the reduced vectors for training. And $\mathbf{Z}_{i,j}^* = \text{Encoder}(G_{i,j}^*, \mathcal{X}_i^1, \mathcal{X}_i^2, \mathbf{w}_{\text{Encoder}}^*)$ is for the $\boldsymbol{\mu}_i \in \Xi$ and time point $t_j$. The training is formulated as

$$\mathbf{w}_{F_r}^* = \underset{\mathbf{w}_{F_r}}{\arg\min} \sum_{i=1}^{N_s} \left\| \mathbf{Z}_{i,j}^* - \big(F_r(\boldsymbol{\mu}_i, \mathbf{Z}_{i,0}^*, \mathbf{w}_{F_r})\big)_j \right\|_2, \tag{16}$$

where $(\mathcal{S})_j$ denotes the $j$-th elements of the ordered set $\mathcal{S}$.

**Remark 8** *The decoupled training of the spatial and temporal models has two advantages. The first is to reduce memory consumption for dynamic training. It allows the attention model to be trained only on the reduced vectors instead of the high-dimensional vectors. More importantly, the gradient of the loss function with respect to neural network weights becomes less memory-consumptive. Thus, the second advantage is that it enables a true auto-regressive training style consistent with the online evaluation instead of teacher forcing [9, 12].*

## D. Evaluations Details

Given a parameter $\boldsymbol{\mu}_i$, initial state $\mathbf{U}_{i,0}$, and relevant quantities $(\mathbf{A}_i, E_i, \mathcal{X}_i^1, \mathcal{X}_i^2)$ from the mesh, the Encoder model (Encoder) firstly generate the initial reduced state vector $(\mathbf{Z}_{i,0})$, together with the parameter, the dynamic model $(F_r)$ directly generates the prediction of reduced vectors and the

Decoder (Decoder) recovers the high-dimensional solution vectors,

$$
\begin{aligned}
G_{i,0} &= (\mathbf{U}, \mathbf{A}_i, E_i), \\
\mathbf{Z}_{i,0} &= \text{Encoder}(G_{i,0}, \mathcal{X}_i^1, \mathcal{X}_i^2, \mathbf{w}_{\text{Encoder}}^*), \\
\{\mathbf{Z}_{i,1}, \dots, \mathbf{Z}_{i,N_t}\} &= F_r(\boldsymbol{\mu}_i, \mathbf{Z}_{i,0}, \mathbf{w}_{F_r}^*), \\
\mathcal{U}_i^{F_r} &= \{\mathbf{U}_{i,1}, \dots, \mathbf{U}_{i,N_t} \mid \mathbf{U}_{i,j} = \text{Decoder}(\mathbf{Z}_{i,j}, E_i, \mathcal{X}_i^1, \mathcal{X}_i^2, \mathbf{w}_{\text{Decoder}}^*)\},
\end{aligned}
\tag{17}
$$

where we obtain the approximation of the numerical solution $\mathcal{U}_i^{F_h} \approx \mathcal{U}_i^{F_r}$ via neural networks with much lower cost.

**Remark 9** *The dynamics evolves directly in the reduced vector space. We can apply the Decoder model to recover the high-dimensional snapshots in a parallel way at the end of the evolution. Alternatively, we can store the reduced vectors using less memory than the original snapshots.*

## E. Hyperparameters of Graph-LED

For the cylinder case, we used 3 GNN layers for the encoder and 3 GNN layers for the decoder. All MLPs are three-layered, with 128 neurons in each level. For the Transformer [47], we used a two-layer model with eight heads and a context length of 32, and all MLPs are three-layer with 128 hidden units. In the backward-facing step case, we have 5 GNN layers for the encoder and 5 GNN layers for the decoder. All MLPs are simply three layers with hidden units of 128. The Transformer is identical to the cylinder case, but the context length was reduced to 8.

# F. Comparison with two baselines

For the cylinder case, we compared our Graph-LED framework with two state-of-the-art methods: MeshGraphNet [9] and NNGraphNet [12, 51]. In the following we have listed a comparison of the results including the Fréchet inception distance (FID), the continuous rank probability score (CRPS), and the errors of velocity, pressure, turbulent kinetic energy, and vorticity. All errors and metrics were computed as averages over a prediction of 100 time steps.

For all scalar error variables, our method outperforms the deep learning baselines with 10 times less error. Similarly for the non-Euclidean processes such as FID and CRPS, here our model also works significantly better. We attribute the significantly better error metrics of Graph-LED to the encoding to a latent space that is then advanced in time using a transformer. Thus, the Graph-architecture is not used for the latent dynamics which is in our test case advantageous due to the highly unstructured mesh that consists of cells with significantly varying size. Moreover, the Transformer architecture can capture long-range dependencies and is thus advantageous for temporal modeling. The poor performance of NNGraphNet is due to instabilities occurring at later prediction time steps as the GNN-based temporal predictions are less stable than the used Transformer. This issue could potentially be addressed by using a more tailored noise injection method. For MeshGraphNet the predictions continued to be stable for the investigated time frame. With regards to speedup compared to a full high-dimensional simulation, all three methods are capable of achieving a high speedup, with Graph-LED and MeshGraphNet outperforming NNGraphNet. The memory required is lowest for Graph-LED as we can store the states using their encoded representation.

To ensure a fair comparison, we used the same GNN architecture that was employed for Graph-LED for MeshGraphNet and the NNGraphNet architecture as published in [12].

| | Graph-LED | MeshGraphNet (w. noise injection) | NNGraphNet (w. noise injection) |
|---|---|---|---|
| $e_u$ | 0.011801029 | 0.36597002 | 8.178367 |
| $e_v$ | 0.024475042 | 0.84774095 | 47.916534 |
| $e_p$ | 0.015479553 | 0.70415086 | 14.178546 |
| $e_{\text{vorticity}}$ | 0.028494475 | 0.79599255 | 25.201374 |
| $e_{\text{TKE}}$ | 0.009462313 | 0.5839962 | 9858.174 |
| FID | 0.007336787 | 673.4697655 | 2860731.926 |
| CRPS | 1.59538563 | 3.716492 | 76.2174 |
| Memory per State | 2MB | 40MB | 40MB |
| Speedup during Predictions | $\approx 900x$ | $\approx 900x$ | $\approx 450x$ |

Table 1: The error is calculated based on the relative root mean squared error (RRMSE), and TKE stands for turbulent kinetic energy [52], FID is for Frechet inception distance based on vorticity [53], CRPS [54] is for continuous ranked probability score based on vorticity. The noise injection during training for both baselines is two percent of the standard deviation of each scalar field [9].

# G. Additional Results

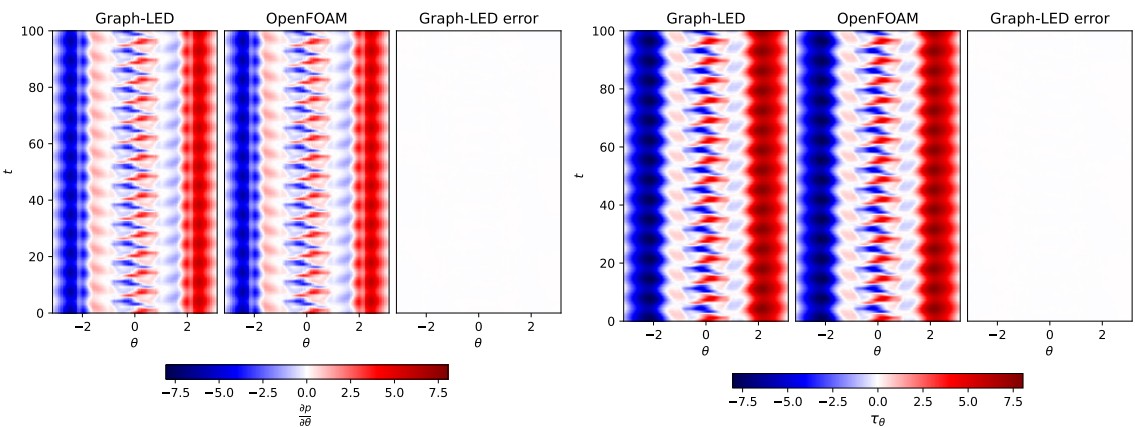

Figure 8: Pressure gradient and wall shear stress ($\tau_\theta = \mu(\frac{\partial(\mathbf{v}\cdot n)}{\partial n})|_\theta$) on the cylinder forecasted over time by Graph-LED and OpenFOAM.

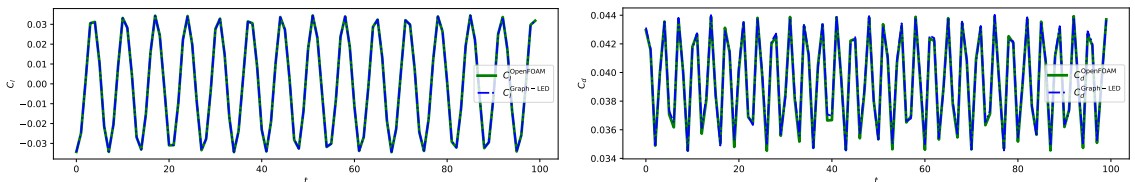

Figure 9: Lift and drag coefficients of the cylinder forecasted over time by Graph-LED and Open-FOAM.

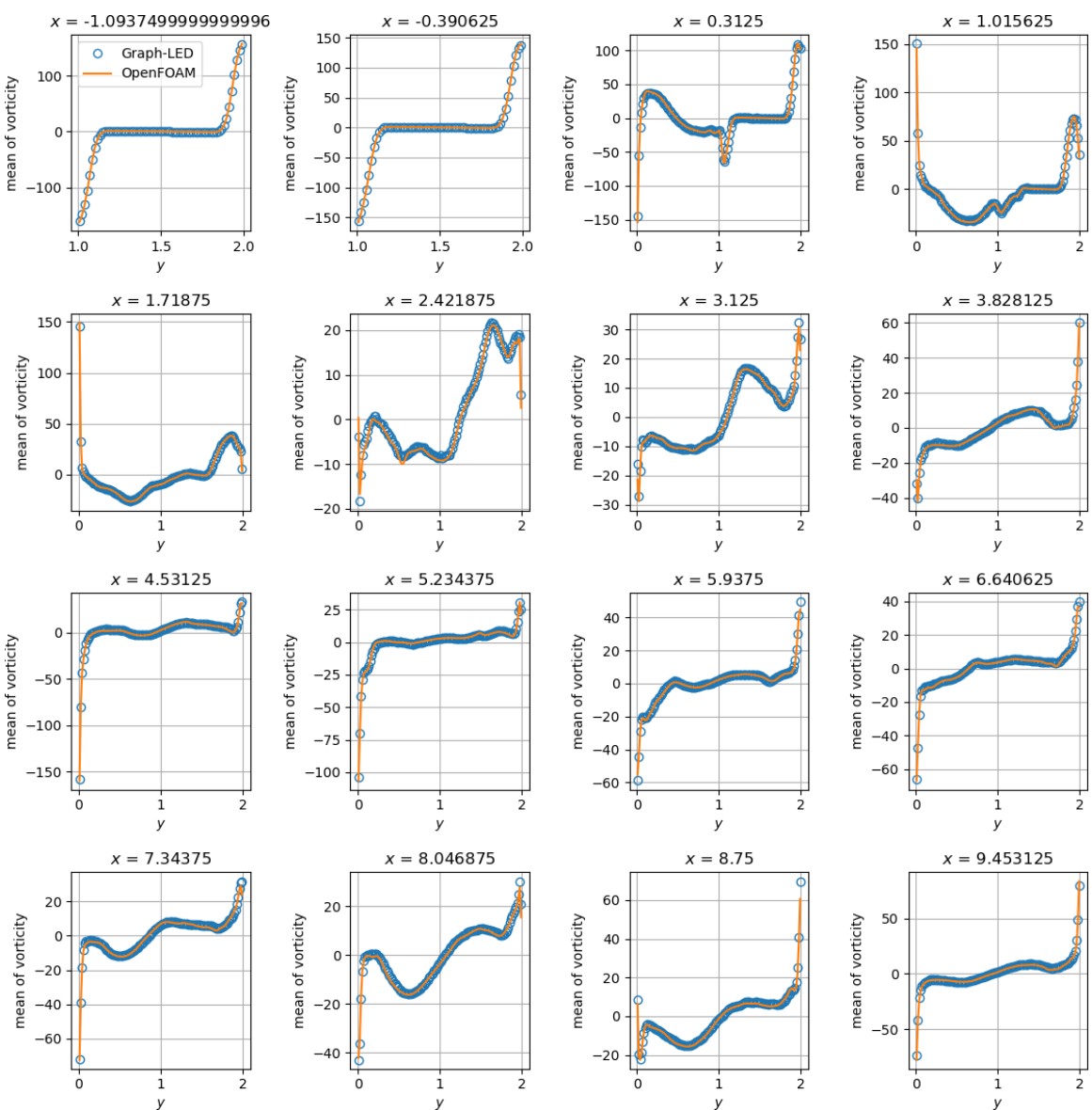

Figure 10: Mean of vorticity over time at different vertical lines over the domain predicted by Open-FOAM and Graph-LED.

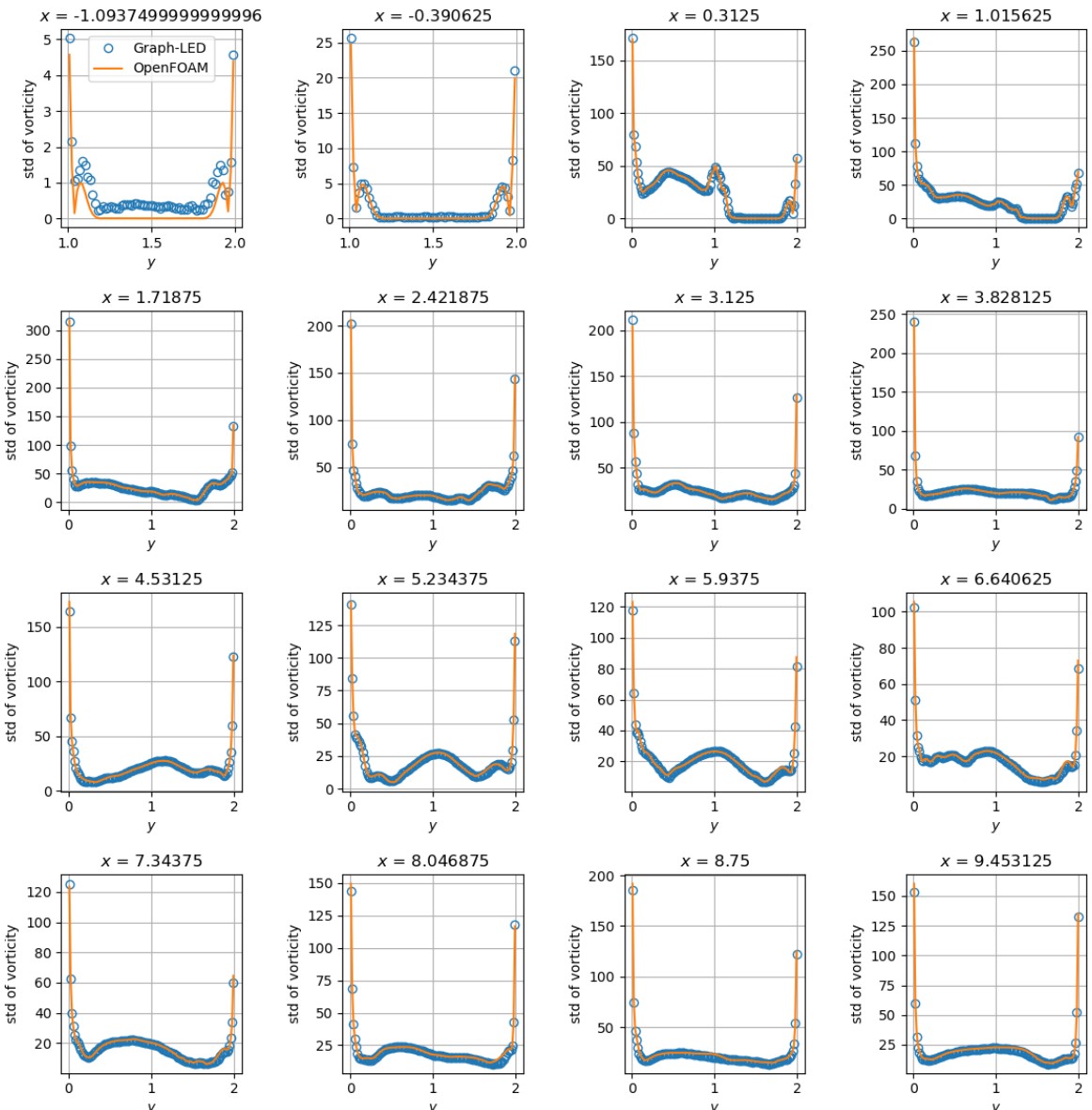

Figure 11: Standard deviation (std) of vorticity over time at different vertical lines over the domain predicted by OpenFOAM and Graph-LED.

