# OpenReview forum: "Learning Effective Dynamics across Spatio-Temporal Scales of Complex Flows"
_CPAL.cc/2025/Proceedings_Track — CPAL 2025 (Proceedings Track) Poster_

### Official Review · Reviewer_qFVW · 2025-01-11
**Promising GNN-based framework for multi-scale fluid flow dynamics, with needs for adding experiments and improving writings**

**Rating:** 8
**Confidence:** 5

**Review:**

CPAL 75

This paper proposes a novel framework combining Graph Neural Networks (GNNs) and an attention-based autoregressive model to learn and predict the effective dynamics of spatio-temporal systems. The framework is applied to fluid dynamics problems, leveraging GNNs for spatial dimension reduction on unstructured meshes and temporal forecasting with an attention mechanism. The results are validated on two complex fluid dynamics scenarios: flow past a cylinder and flow over a backward-facing step.

Pros:
1. Innovative approach
2. efficient dimension reduction
3. impressive speed-up
4. strong experiment results

Cons:
Major:
1. Please consider adding baseline for comparison. Authors introduced teacher-forcing methods and sequence neural networks in the introduction section and stated the limitations of these methods, respectively. Adding comparison and discussions (or ablation studies), e.g, error accumulation for long trajectories, computation/memory cost for each type of methods and the proposed frameworks would significantly improve the paper.
2. Personally, I believe the writings can be improved in many ways. Please see minor.

Minor:
1. Please consider designing a name for your framework to make it easier for readers to memorize and cite.
2. Authors mentioned low-dimensional modeling in the introduction, however, the proposed method does not ????????
3. GNN has already been used for dimension reduction. I would recommend authors to cite some of those works.
4. Please consider improve the information density and look of Figure 1. For example, use the overview of the framework as Figure 1 and add another figure showing graphs, and the design of your GNN (as a visualization of details in Section 2).
5. Please consider add a short paragraph between line 89 and 90 for overview of the section 2.
6. As authors introduce the NS equations in Section 2.1, I would suggest adding vorticity definitions there, too. Because it's the main results visualized for your experiment.
7. Line 91, abbreviation of PDE already defined in line 36. Line 168, abbreviation of MLP already defined in line 139. No need to re-define. Same in Line 170 for LNM and line 171 and line 269 for GNN.
8. Line 109, I thought the problem defined in section 2.1 was a 2D problem, so there might be no 'shared face' of adjacent cells, but an edge (though the term 'edge' is defined for $e$)?
9. Line 113, I guess the size of matrix U should be $N_u \times N$? Same in line 125.
10. Line 116, I guess N_E should be 1/2 of that, since the matrix A is a symmetric matrix, assuming the diagonal of matrix A is zero (also not specified in the text)?
11. In Section 2.3, the first sentence states 'xxx are effective representations of unstructured meshes'. However, the word 'unstructured' did not appear in the section 2.2, where graph representations are defined. Improvement of writings are encouraged.
12. Please consider adding the letter representations into your figure captions. For example, in Figure 2, saying 'Given a high-dimensional initial state U_0, the encoder... ' might improve readability.
13. Line 156, $N_1$ not defined before and not used later.
14. Line 158, is the X_2 intentionally starting with x_2?
15. Line 160, I would recommend talking more about the nearest-neighbor here. Since it is one of the cores of your dimension reduction techniques.
16. Line 173, there should be a space before 'Given'.
17. Line 197, missing a period in the end. Same in line 229.
18. Eq (11), the $\cdot$ operation, which should be the inner product between two vectors, is not specified.
19. Not enough implementation details are provided for experiments. Maybe adding a table or subsection for framework details, like hyperparameters etc can be helpful.

---

### Official Review · Reviewer_PAxN · 2025-01-11
**An effective framework, but more rigorous quantative experiments would make the claim more solid**

**Rating:** 5
**Confidence:** 4

**Review:**

The paper builds an effective robust framework for flow simulating complex fluid flows. It achieves computational efficiency via dimensionality reduction at the spatial dimensions and an attention-based autoregressive model along the temporal dimension. Overall, the experimental results look good. However, I think more rigorous experimental comparisons should be conducted to quantitatively show the effectiveness of the proposed method, in addition to the qualitative results. For example:

- One advantage of the proposed method is said to be its computational efficiency. Then can it be justified with experiments? e.g. what are the memory consumption / time consumption v.s. accuracy for different settings, such as with or without the dimensionality reduction.
- Currently, the reported error maps are just the errors between OpenFOAM and the proposed method. But to my feeling, if there's only one error map, it's very hard to tell whether it's small or large as the color scales very much depend on the error bar color maps. Could there be more quantitative explanations of why these errors should be interpreted as small errors?

Overall, simply looking at the qualitative results, I tend to agree that the proposed framework is effective. But I think to make it a more solid claim it'd be better to have more rigorous quantitative analysis.

---

### Official Review · Reviewer_BrKJ · 2025-01-17
**Integrating GNNs and attention based autoregressive models for effective, multi-scale fluid dynamics modeling**

**Rating:** 7
**Confidence:** 3

**Review:**

Fluid dynamics modeling is inherently computationally intensive, and traditional methods often struggle to accurately capture the multiscale spatiotemporal dynamics within unstructured mesh data. This paper introduces a novel framework known as Graph-LED (Learning Effective Dynamics), which integrates Graph Neural Networks (GNNs) with attention-based autoregressive models.

The main ideas include
- Using GNNs for efficient spatial dimension reduction on unstructured meshes.
- Employing temporal attention mechanisms for dynamic forecasting
- Integrating these components to create a robust and efficient surrogate model for complex fluid dynamics simulations.

The authors evaluate this approach through experiments conducted on two case studies:
- Flow past a cylinder at a Reynolds number of 696
- Turbulent flow over a backward-facing step at a Reynolds number of 5000.

**Strengths:**
- The proposed Graph-LED approach demonstrates a well-motivated integration of graph neural networks and sequential networks, effectively leveraging their respective strengths.
- The authors show that fluid fields generated by Graph-LED are comparable in quality to those produced by traditional CFD simulations, while achieving significant efficiency gains. Notably, they report speedups of 900x for cylinder flow simulations and 100x for backward-facing step simulations.
- The paper is generally well organized and clearly written.

**Weaknesses:**
- Although the paper demonstrates substantial speedups over traditional CFD methods, it does not provide a comparison with other deep learning-based approaches. It would be interesting to see how its performance compares against more advanced baselines in the field.

---

### Meta-Review · Area_Chair_owpg · 2025-02-02

**Recommendation:** Accept (Poster)
**Confidence:** 5

**Metareview:**

This paper combines Graph Neural Networks and an attention-based autoregressive model to learn and predict the dynamics of spatio-temporal systems.
The reviewers valued the proposed approach as novel and effective. Some reviewers valued the paper's presentation and numerical experiments. The main criticism was the lack of quantitative metrics to assess the model's performance against baselines. However, the authors addressed that issue in the rebuttal.

---

### Decision · Program_Chairs · 2025-02-11

Accept (Poster)